# The Effectiveness of Mobility Restrictions on Controlling the Spread of COVID-19 in a Resistant Population

**DOI:** 10.3390/ijerph20075343

**Published:** 2023-03-31

**Authors:** Dina Albassam, Mariam Nouh, Anette Hosoi

**Affiliations:** 1King Abdulaziz City for Science and Technology (KACST), Riyadh 11442, Saudi Arabia; dalbassam@kacst.edu.sa; 2Institute for Data, System and Society (IDSS), Massachusetts Institute of Technology (MIT), Cambridge, MA 02139, USA; peko@mit.edu

**Keywords:** COVID-19, Delta-variant, fraction of resistance, human mobility, pandemic, Pearson correlation method

## Abstract

Human mobility plays an important role in the spread of COVID-19. Given this knowledge, countries implemented mobility-restricting policies. Concomitantly, as the pandemic progressed, population resistance to the virus increased via natural immunity and vaccination. We address the question: “What is the impact of mobility-restricting measures on a resistant population?” We consider two factors: different types of points of interest (POIs)—including transit stations, groceries and pharmacies, retail and recreation, workplaces, and parks—and the emergence of the Delta variant. We studied a group of 14 countries and estimated COVID-19 transmission based on the type of POI, the fraction of population resistance, and the presence of the Delta variant using a Pearson correlation between mobility and the growth rate of cases. We find that retail and recreation venues, transit stations, and workplaces are the POIs that benefit the most from mobility restrictions, mainly if the fraction of the population with resistance is below 25–30%. Groceries and pharmacies may benefit from mobility restrictions when the population resistance fraction is low, whereas in parks, there is little advantage to mobility-restricting measures. These results are consistent for both the original strain and the Delta variant; Omicron data were not included in this work.

## 1. Introduction

In December 2019, a new coronavirus (COVID-19) was discovered in Wuhan, China. Subsequently, it rapidly spread throughout the world, leading to 6.4 million deaths worldwide as of August 2022 and precipitating a global pandemic [1,2]. It has now been established that COVID-19 is primarily transmitted through in-person interactions between people. Therefore, human behavior and human mobility play an important role in determining how the virus spreads [3,4]. In the absence of vaccination, non-pharmaceutical interventions (NPIs) have been applied by many countries to control the spread of the disease. One widely adopted NPI was human mobility reduction [5,6,7] implemented via the closure of public spaces.

Many excellent studies have been conducted on the effect of human mobility in the evolution of the COVID-19 pandemic. Prior work can be broadly classified into two categories: foundational and policy studies. In many foundational studies, researchers seek to find correlations between mobility and the evolution of the pandemic at country, county, and point of interest (POI) levels. One such study [8] investigated the spatiotemporal association between mobility and infections in US counties. Researchers used mobile device data to capture the mobility flow within and into each county and compared mobility trends with COVID-19 case counts using a dynamic time-warping method. They found that the relationship between mobility and infection rates varies both geographically and temporally. A similar study [9], used mobile device data to find the association between mobility and case counts for US counties. Their analysis shows a positive relationship between mobility and the number of cases and suggests that this relationship is stronger in partially reopened regions. In [10], the authors assessed the correlation between mobility and the number of new cases in different Portuguese districts. They found that mobility in retail and recreation, grocery and pharmacy, and transport station POIs exhibited a higher correlation with case counts than in parks and workplaces.

Beyond correlation, some studies expanded their analysis to evaluate the causal factors behind increased rates of transmission. For example, Refs. [11,12] analyzed the effect of temperature on the transmission of COVID-19. Researchers in [11] used a directed acyclic graph (DAG), a graphical representation of the causal effects that may lead to newly reported cases of COVID-19 they found an increase in temperature and high mobility (in pharmacies and groceries), lead to lower case counts. On the other hand, high mobility (in retail and recreation POIs), and rainy days, lead to higher case counts. While in [12], their estimated results showed that mobility habits, along with daily tests and environmental variables, such as temperature, play a role in explaining the rate of COVID-19 cases. In addition, some studies focused on quantifying the time lag between mobility and COVID-19 cases as in [13]. In that study, researchers combined the mobility index of 80 cities in China along with new case counts and used an autoregressive model to estimate the lag. As a result, they found that mobility is strongly correlated with cases with a lag of 10 days.

The second type of studies focused on policymaking, e.g., testing different mobility interventions to find optimal mobility reductions that balance the cost of viral spread with the economic cost associated with lockdowns, as well as implementing prediction models to advise policymakers. Several studies [13,14,15,16] sought to understand how the reduction in mobility affects the spread of COVID-19 cases across different POIs. In [13], researchers used Google mobility data and measured a correlation with the effective reproduction rate $R_{t}$. That study reveals that staying at home is effective at reducing $R_{t}$, time spent at parks has little effect, while reducing mobility in other POIs has larger positive effects. In [15,16], researchers showed that mobility reduction of up to 40% in transit stations and retail and recreation venues decreased the number of cases and appeared to effectively “flatten the curve”. Furthermore, Refs. [17,18] examined the effect of the reduction of mobility on the number of cases and deaths. In [17], researchers found a consistent pattern of a sharp reduction in deaths after mobility is reduced. Other groups implemented prediction models [19,20,21,22,23,24,25] to estimate the effects of mobility reduction and predict the number of cases and deaths. These models were implemented with varying levels of complexity; for instance, [19,20] added additional variables, including (in [19]) meteorological variables, such as temperature, humidity, and rainfall, along with the correlation between mobility and COVID-19 case counts. In [20], the researchers included several factors such as income, health indicators associated with Asthma, percentage of people staying at home, and testing infrastructure.

None of the studies described above accounted for the fraction of the population with resistance. However, as the pandemic progressed, population resistance increased via natural immunity and vaccination. Many studies were conducted to measure the efficacy of vaccines either in terms of impact on transmission or the number of deaths, but the effects of mobility in these studies are typically either neglected (e.g., the studies considered populations of vaccinated and unvaccinated people with similar mobility patterns) or treated as a confounding variable. One such study [26] used a deep learning approach to simulate vaccination rates and time to reach herd immunity based on the data from eight countries in Asia (many subsequent studies have shown that herd immunity is no longer in our grasp), whereas [27,28] evaluated the impact of vaccination in controlling the pandemic (e.g., reducing the number of incidences and deaths) via an agent-based model. Furthermore, Ref. [29] assessed the association between vaccination and death rates in the US using a regression-based approach and as a result, found that vaccination helped in reducing the death rates in different states in the US.

Taking into account the impact of vaccination and mobility on the transmission, Ref. [30] examined the vaccination and mobility behavior in controlling the pandemic using structural equation modeling; they found that vaccination slows down the spread of COVID-19 in regions where the vaccination is negatively correlated with mobility and vice versa for the regions that have a positive correlation between vaccination and mobility.

Although the previous work has investigated the correlation between mobility and the evolution of the pandemic, including the effect of vaccination as in [30], they did not consider the effect of population resistance (both natural immunity and vaccination) on this correlation. Therefore, the focus of our study is to analyze the correlation between mobility and COVID-19 transmission across different points of interest (POIs) while taking into account population resistance (vaccination—natural immunity (% of the population who recovered from COVID-19)) and the emergence of the Delta variant.

To achieve our goal, we will exploit the Our world in Data COVID-19 case count dataset, along with the Google mobility dataset. We combine these datasets to test the correlation between mobility and the rate of viral spread while accounting for four additional factors: points of interest (POIs)—including retail and recreation venues, transit stations, parks, groceries and pharmacies, and workplaces; the emergence of the Delta variant; the fraction of the population that has been vaccinated; and the fraction of the population with natural resistance (i.e., those that have recovered from a COVID-19 infection). Note that if there is no correlation between mobility and COVID-19 case counts, then mobility-restricting mitigation measures will have little impact on controlling the spread of the virus.

The remainder of this paper is organized as follows. Section 2 introduces an overview of the dataset and data preprocessing. Section 3 shows our methodology. Section 4 and Section 5 show the results of our analysis, along with a discussion. Finally, conclusions are drawn in Section 6.

## 2. Datasets

The first step in the analysis was to collect and preprocess the data; this section provides an overview of the datasets we used for the study and our country selection process. Our data were pulled from the three datasets shown in Table 1 with a selected time frame of Feb 2020–Jul 2021.

### 2.1. COVID-19 Dataset

Our World in Data [31] is maintained by a non-profit organization that includes thousands of researchers from around the world. One of the richest datasets they collected during the pandemic is the COVID-19 dataset. This dataset captures epidemic measurements daily (e.g., new cases, new deaths, vaccinations, etc.) along with demographics (e.g., median age) and country-related metrics (e.g., human development index). It covers 231 countries and includes 60 variables. The data collection began in Jan 2020, and, to date, the dataset has been updated every day.

### 2.2. Google Mobility Dataset

Google provides a publicly available mobility dataset related to the pandemic [32], in which it measures daily visitor numbers to specific POIs as a mobility rate (MR). POIs include transit stations, groceries and pharmacies, retail and recreation venues, workplaces, and parks. Their measurement is based on computing the relative change in visitors from a baseline before the pandemic and covers 123 countries. The data collection began in Feb 2020, and it has been updated daily to date.

### 2.3. COVID-19 Variants

In addition to the COVID-19 dataset described above, Our World in Data shared a COVID-19 variants dataset [33], which is sourced from GISAID [34]. The dataset periodically captures the number of sequenced samples that fall under a specific variant name. It covers 110 countries. Data collection began in May 2020, and, to date, it has been updated every two weeks.

### 2.4. Country Selection

Throughout the pandemic, there have been many inconsistencies and uncertainties surrounding testing and accurately reporting the number of COVID-19 cases in different countries. Given this uncertainty, we endeavored to restrict our study to countries with relatively reliable reported data. Our selection process followed the steps shown in Figure 1. First, we found the intersected countries from the three datasets. Then, to get a rough sense of which countries were detecting a large fraction of cases, we found, for each country, the largest ratio of detected cases ($C_{new}$) to the total population (*p*):(1)$$R=\frac{max(C_{new})}{p}.$$

If each country was hit with a wave of cases at some point during the pandemic, the higher this ratio is, the more likely it is that country is reliably detecting and reporting cases; i.e., if a country reports that this ratio is near zero, given the highly transmissible nature of the virus, it is more likely that they are undercounting rather than that they have no COVID-19 cases. There are, of course, exceptions to this; for example, early on in the pandemic, New Zealand enforced draconian measures to prevent the spread and largely kept the virus contained. Countries were then sorted by this ratio, and the highest 50 ratios were selected as regions where COVID-19 is likely to be relatively reliably detected and reported. In order to focus on the effects of population resistance, out of those 50, we selected the countries that achieved 60% population resistance or more via vaccination or recovery from past infections by July 2021. This left us with our 14 target countries (Argentina, Canada, United States, United Kingdom, Italy, Austria, Ireland, Czech Republic, France, Uruguay, Slovenia, Israel, Switzerland, Luxembourg), as shown in Figure 2.

## 3. Methodology

We divided our methodology into three steps: Daily Measurements, Correlations and Population Resistance Ranges, and Delta Variant Estimates. All the code has been written from scratch using Python.

### 3.1. Daily Measurements

In this section, our main aim is to compute the following daily variables: mobility rate (for each of the five POIs), growth rate ($GR_{j}^{t}$) of cases, and the fraction of the population with resistance (FR), as shown in Figure 3. For the mobility rate, we take values directly from the Google Mobility dataset. Other variables were calculated as follows.

To estimate the growth rate of cases, we apply Formula (Equation 2) from [21] with an 11-day lag between mobility measurements and the instantaneous temporal derivative of the number of cases based on the optimal lag findings in [35]:(2)$$GR_{j}^{t}=\frac{\log(\sum_{t-2}^{t}\frac{c_{j}^{t}}{3})}{\log(\sum_{t-6}^{t}\frac{c_{j}^{t}}{7})}.$$

The growth rate of cases ($GR_{j}^{t}$) for a specific country *j* and a given day *t* is calculated as the logarithmic rate of change for the new cases over the previous three days relative to the logarithmic rate of the new cases over the previous week. Here $c_{j}^{t}$ is the number of new cases in country *j* on day *t*.

To estimate the fraction of the population with resistance for a specific country on a given day, we combined the fraction of people with natural immunity (i.e., those that have already contracted COVID-19 and recovered) with the fraction of vaccinated people: (3)$$\begin{array}{ccc} FR_{T} & = & FR_{c}+FR_{v}\left(1-FR_{c}\right) \end{array}$$(4)$$\begin{array}{ccc} FR_{c} & = & \frac{m\cdot (C_{Tot})}{p} \end{array}$$(5)$$\begin{array}{ccc} FR_{v} & = & \frac{V_{Tot}}{p} \end{array}$$
where the total fraction of people with some form of resistance ($FR_{T}$) depends on the fraction of cases ($FR_{c}$) to date, which is estimated by scaling the total number of reported cases ($C_{Tot}$) by a multiplier (*m*) for each country, as shown in Table 2. These multipliers *m*, which account for under-reporting, were taken from numerous studies in the literature. This approach assumes that natural resistance has not faded, which may be an acceptable assumption during the initial stages of the pandemic. If this analysis were to be repeated, we would recommend restricting the window of cases that lead to natural resistance to account for the fact that (a) it is likely that the pandemic has outlasted the typical duration of resistance, as evidenced by the enormous numbers of reinfections observed in the past year, and (b) it appears that the Omicron variants are particularly good at evading our defenses and reinfection may occur on fairly short time scales. The fraction with natural resistance ($FR_{c}$) is added to the fraction of the population that has been vaccinated ($FR_{v}$), which is estimated in Equation (5). To avoid double counting people who both contracted COVID-19 and were also vaccinated, $FR_{v}$ is scaled by (1$-FR_{c}$).

### 3.2. Correlations and Population Resistance Ranges

After estimating the daily variables, we bin the time series of the countries into ranges based on $FR_{T}$. This results in different periods of time for each country in which the percentage of the population with resistance is similar. To achieve this, we followed the process in Figure 4, where we first considered each of the countries and split the time series into periods based on the total fraction of resistance $FR_{T}$: we select the first bin of $FR_{T}$ to lie between 0–10%, the second between 10–20%, and so on up to 60–70%. Then for each period, we calculated the correlation between the time-lagged google mobility score and the growth rate of cases ($GR_{j}^{t}$) for each one of the POIs. We used a Pearson correlation, where 1 indicates a strong positive correlation, 0 is no correlation, and −1 a strong negative correlation.

### 3.3. Delta Variant Estimates

To determine how the emergence of the Delta variant affects these correlations, we calculated the fraction of Delta for each period of $FR_{T}$ and identified periods with high fractions of Delta and low fractions of Delta (<10%) (Figure 5). This was accomplished by first filtering the COVID-19 variants dataset based on country and variant. The start and end dates for each $FR_{T}$ period were identified, and 14 days of lag were added to each (since the sequenced samples were only collected every two weeks). Then the nearest dates from the COVID-19 variants dataset were selected to represent the relevant period. The following formula was used to estimate the fraction of the Delta variant ($F_{\Delta }$):(6)$$F_{\Delta }=\frac{n_{\Delta }}{n_{Tot}}$$
where ($F_{\Delta }$) equals the sequenced samples of the Delta variant ($n_{\Delta }$) divided by all the sequenced samples during that period ($n_{Tot}$).

Finally, we aggregated the correlation coefficients for all countries and for each of the POIs and $FR_{T}$ periods; we consider two transmission trajectories, one which only includes data for which Delta < 10% (Pre-Delta) and the other one which includes all of the data (post-Delta).

## 4. Results

### 4.1. Correlation between Mobility and COVID-19 Cases

The histogram of the measured correlations between the growth rate of cases and mobility across all 14 countries and for all of POIs is shown in Figure 6. The correlation ranges from −0.67 to 0.66, with most of the data in the range [−0.29, 0.28]. At this granularity, the data suggests that there is no real correlation between mobility and growth rate, i.e., on average, modifying mobility patterns does not affect the number of cases. However, this coarse-grained approach masks the effect of population resistance, which will be revealed in the following sections.

### 4.2. The Effect of Population Resistance on COVID-19 Transmission

As the pandemic progresses, the fraction of the population with resistance to the virus increases either via vaccination or by contracting COVID-19 and recovering. To get a sense of the impact of resistance on the correlation of growth rate with mobility, consider the following two extreme cases. In a naive population in which no one has any resistance to the virus, every contact represents a potentially new infection; in this scenario, spending time in POIs that increase the number of contacts in the population (e.g., transit stations) is likely to lead to more cases. Hence, in a naive population, we expect mobility in high-contact POIs to be strongly correlated with the growth rate of cases. At the other extreme, if the bulk of the population is resistant to the virus, most of the contacts in a given POI are unlikely to lead to transmission. This suggests that in a strongly resistant population, mobility in high-contact POIs will be only weakly correlated with growth rate, if at all.

To test this, we binned the data by the fraction of the population with resistance, $FR_{T}$ (which is estimated from both vaccination rate and previous infections as described above). The results of this exercise for recreation and retail POIs are shown in Figure 7. Each point represents the mean correlation coefficient (averaged across all countries during the relevant time periods) between the growth rate of COVID-19 cases and mobility. The bars capture 75% of the data; outliers have been removed to declutter the visualization and highlight trends. The standard error in the mean is 0.056, as shown by the grey region in the plot. As expected, when only a small fraction of the population is resistant (approximately 0–25%, as shown by the orange points in this instance), there is a clear correlation between mobility and the growth rate of cases, suggesting that restricting access to recreation and retail establishments early on in the pandemic may be beneficial in curtailing the number of transmissions. However, as resistance in the population grows, the correlation between mobility and the growth rate of cases becomes successively weaker.

Similar plots showing the effect of population resistance on the four other POIs—transit stations, workplaces, groceries and pharmacies, and parks—are shown in Figure 8. All POIs show similar trends with clear correlations between mobility and growth rate of cases when the fraction of the population with resistance is low; as resistance increases, the correlations steadily decrease, becoming negligible at different critical values of $FR_{T}$.

Transit stations reveal a similar trajectory to retail and recreation, as shown at the top of Figure 8, where for $FR_{T}$ between 0–25%, there is a significant mean correlation (>0.056) between mobility and growth rate. Then, as $FR_{T}$ increases above 25%, the correlation steadily decreases; workplaces show a similar trend as well. For groceries and pharmacies and parks, the correlation is not as strong as with transit stations, retail and recreation, or workplace venues. As shown in Figure 8, there is a clear correlation when $FR_{T}<10$%. After that, correlation means hover near zero, suggesting that restricting mobility may be helpful in some cases and not in others; whether or not this is the case likely depends on other mitigation factors that are in place.

The mean correlation thresholds used to distinguish between correlated (orange), uncorrelated (light blue), and slight negative correlation (dark blue) in this analysis, along with their restriction recommendations, are shown in Table 3. Here we have taken a conservative approach in which any correlation mean greater than the standard error in the mean, >0.056, is flagged as positively correlated (orange), hence restricting mobility during that period may be helpful. On the other hand, correlation means that are <−0.056 are negatively correlated and are marked as dark blue. The correlation means between −0.056 and 0.056 represent a regime in which no significant correlation was measured; hence, within the sensitivity of the measurement, mobility restrictions are unlikely to have a significant impact on the growth rate of cases; these points are indicated with light blue markers in Figure 8.

## 5. Discussion

### 5.1. Implications for Mobility-Restricting Policies

All of our data show a clear and consistent trend: in a naive population, mobility is correlated with the growth rate of cases; as the population gains resistance, this correlation declines to zero when $FR_{T}$ reaches a critical value. Once this critical value of $FR_{T}$ is attained and there is no correlation between mobility and growth rate (as is the case for the blue points in Figure 7 and Figure 8), reducing mobility by closing POIs can have little effect on the rate of transmission. It has been well-established that shutting down POIs may incur serious economic and social costs. Hence it is a useful exercise to consider whether the trade-off is worth it. All of the data shown in Figure 7 and Figure 8 suggest that mobility restrictions may be effective early in the pandemic but gradually lose their potency as the population gains resistance via vaccination or recovery and, consequently, the connection between mobility and transmission weakens. Here we have highlighted values of $FR_{T}$ (in orange) for which policies that restrict mobility may have a positive impact on slowing the spread of COVID-19. Similarly, the blue point indicates values of $FR_{T}$ for which mobility restrictions are unlikely to have a significant impact on transmission.

These blue and orange groupings are summarized in Table 3. The data suggest that retail and recreation venues, transit stations, and workplaces are the POIs that stand to benefit the most from mobility-restricting measures and are likely to continue to benefit beyond the initial stages of the pandemic (i.e., until FR grows above 25–30% as shown in Table 4). We speculate that this may arise because these POIs share common underlying characteristics that facilitate transmission (e.g., indoors, crowded, and relatively long residence times).

The second group of POIs that may benefit from mobility restrictions early on in the pandemic (but returns may be less pronounced beyond the initial stages) are groceries and pharmacies. This again shows a clear trend that in a naive population (up to 10% resistant) mobility and growth rate of cases are clearly correlated; hence restricting access or controlling density may be beneficial. We speculate that groceries and pharmacies may fare better than retail and recreation and workplaces because some subset of the patrons’ “pick and go” reduces residence time.

However, it is important to note that these critical values—for all of the venues discussed above—are averaged, and the critical value of $FR_{T}$ of a particular venue likely depends on the POI’s size, local mitigation strategies, etc. While this study provides some general “rules of thumb”, as with any policy, it is important to take a deeper dive to understand local confounding factors to determine which POIs to keep open and which ones to restrict.

Finally, the data show very little correlation between mobility and growth rate in parks for almost all values of $FR_{T}$. For $FR_{T}>$ 5–10%, the correlation was negligible, suggesting that for COVID-19, keeping parks open, which provides an outdoor space for people to socialize, is unlikely to lead to increased transmission (and may prove beneficial if the presence of readily available outdoor spaces incentives people to move out of more transmissive environments).

### 5.2. The Effect of Delta Variant on COVID-19 Transmission

One challenge with our approach is that the fraction of the population with resistance does not occur as a randomized sample in time: resistance increases as the pandemic progresses. This introduces the possibility that the correlations we measure are not due to resistance but rather due to the different variants that evolve as the pandemic progresses. During the time period we consider in this study, the dominant variants were the initial strain(s) and the Delta variant, which proved to be both more transmissible and more deadly than the original strains. To account for the emergence of Delta, we ran the same analysis but removed all points in which Delta accounted for more than 10% of the detected cases. In Figure 7 and Figure 8, the solid line corresponds to this analysis, i.e., the original strains without Delta (<10% Delta), and the dashed line indicates correlations, including Delta variants. All of our analyses were performed before the emergence of the Omicron variants. The results suggest that the emergence of the Delta variant does not impact any of our conclusions.

## 6. Conclusions

In this study, we analyzed the impact of mobility-restricting mitigation measures on a resistant population in which we considered two factors: first, the effect of resistance for different points of interest (POIs), including transit stations, groceries and pharmacies, retail and recreation venues, workplaces, and parks; second, the effect of resistance before and after the emergence of the Delta variant. We studied a group of 14 countries with relatively reliable COVID-19 case counts—in which we binned the data by the fraction of the population with resistance and the presence of the Delta variant—using a Pearson correlation between mobility and cases. We found that mobility restrictions on retail and recreation venues, transit stations, and workplaces are most likely to be beneficial in controlling the spread of the virus, particularly if the fraction of the population with resistance is below 25–30%. In some cases, groceries and pharmacies may benefit from mobility restrictions if the fraction of the population with resistance is below 10%. Whereas in parks, there is little advantage to mobility-restricting measures.

One limitation of this study is the uncertainties in data collection during COVID-19. However, we made an effort to address this problem by using the calculation for country selection described in Section 2.4, where we chose the 14 most reliable countries for the study. On the other hand, this work’s strengths are its global perspective and broad insight. Furthermore, as policy implications, these insights can be used later on for any infectious diseases similar to COVID-19, as discussed in Section 5.1. Knowing which POIs to restrict more while taking the resistance threshold into consideration can help to prevent the spread of the disease and save lives from earlier stages.

In future work, we aim to study the effect of population resistance on both the death rate and the number of ICU patients. In addition, it may be interesting to compare the effect of different types of resistance (natural resistance versus vaccine versus a combination of the two). In addition, as more data becomes available, micro-scale analyses become feasible when one includes different districts and places within cities.

## Figures and Tables

**Figure 1 ijerph-20-05343-f001:**
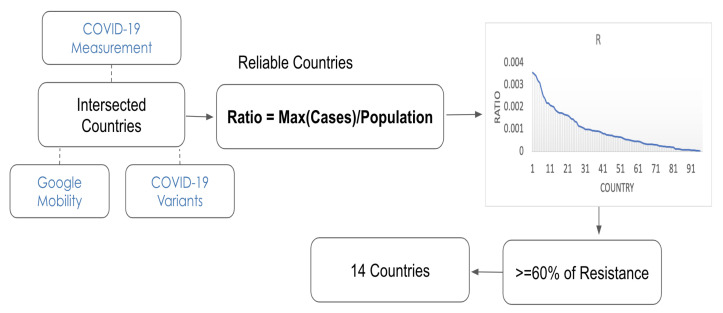
Flowchart of the country selection process based on the largest ratio of detected cases to the population size. Countries that exceed this threshold are included in our dataset if 60% of the population or more have acquired some form of resistance (via vaccination or prior infection) by July 2021.

**Figure 2 ijerph-20-05343-f002:**
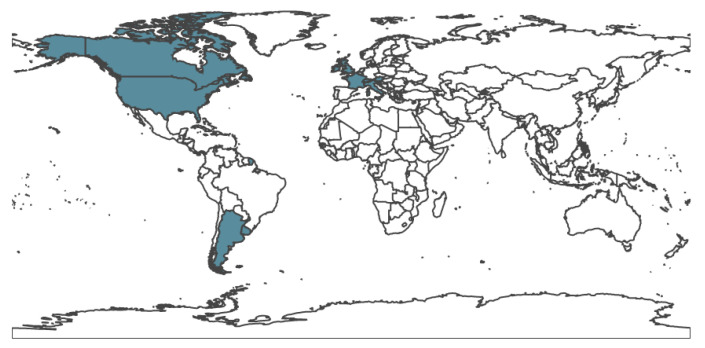
Countries with relatively reliable COVID-19 case counts and which achieved high resistance to the virus (either by vaccination or by prior infections) by July 2021 are colored in teal. These 14 countries were used in our analysis.

**Figure 3 ijerph-20-05343-f003:**
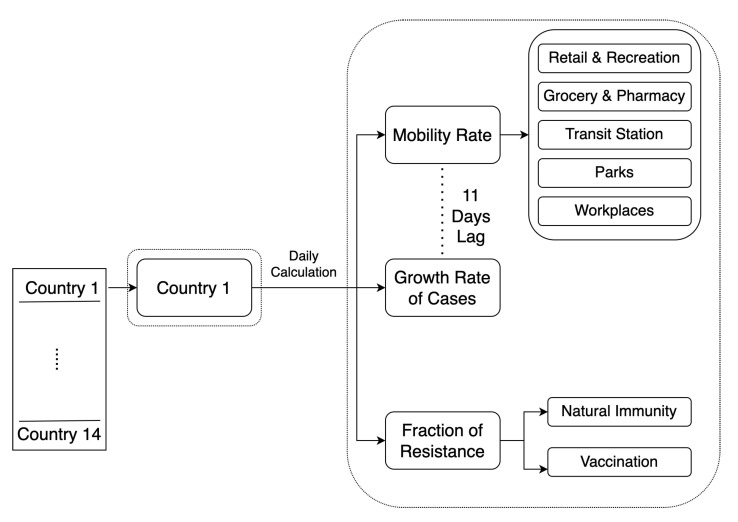
Flowchart showing the daily measurements calculated for each of the 14 countries, including mobility rate for the five POIs, the growth rate of cases with 11 days lag, and the fraction of resistance based on natural immunity and vaccination.

**Figure 4 ijerph-20-05343-f004:**
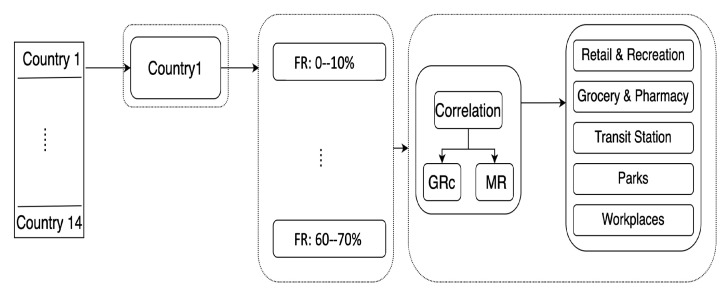
Flowchart showing the splitting of the fraction of resistance periods for each of the 14 countries and measuring the Pearson correlation between MR and GRc for all of the 5 POIs.

**Figure 5 ijerph-20-05343-f005:**
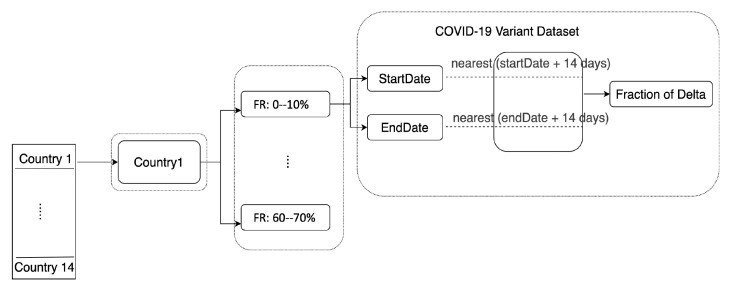
Flowchart showing the calculation of the fraction of Delta variant for each fraction of resistance period across all of the 14 countries.

**Figure 6 ijerph-20-05343-f006:**
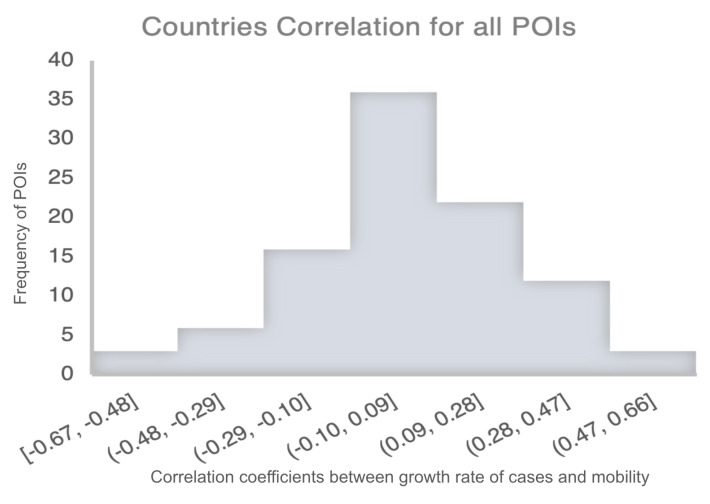
Histogram showing the frequency of the measured correlation coefficients between the growth rate of cases and mobility for all the POIs in all 14 countries.

**Figure 7 ijerph-20-05343-f007:**
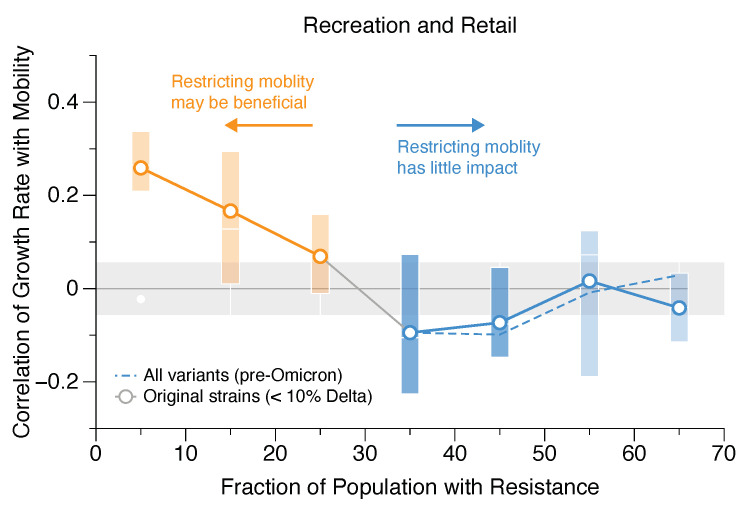
Correlation between COVID-19 transmission rates and mobility in recreation and retail establishments as a function of the fraction of the population with resistance. The solid line only includes pre-Delta variants; the dashed line represents all variants, including Delta. Points in which there is a clear correlation are orange; points where the correlation is indistinguishable from zero (suggesting that mobility-restricting measures are no longer effective) are blue.

**Figure 8 ijerph-20-05343-f008:**
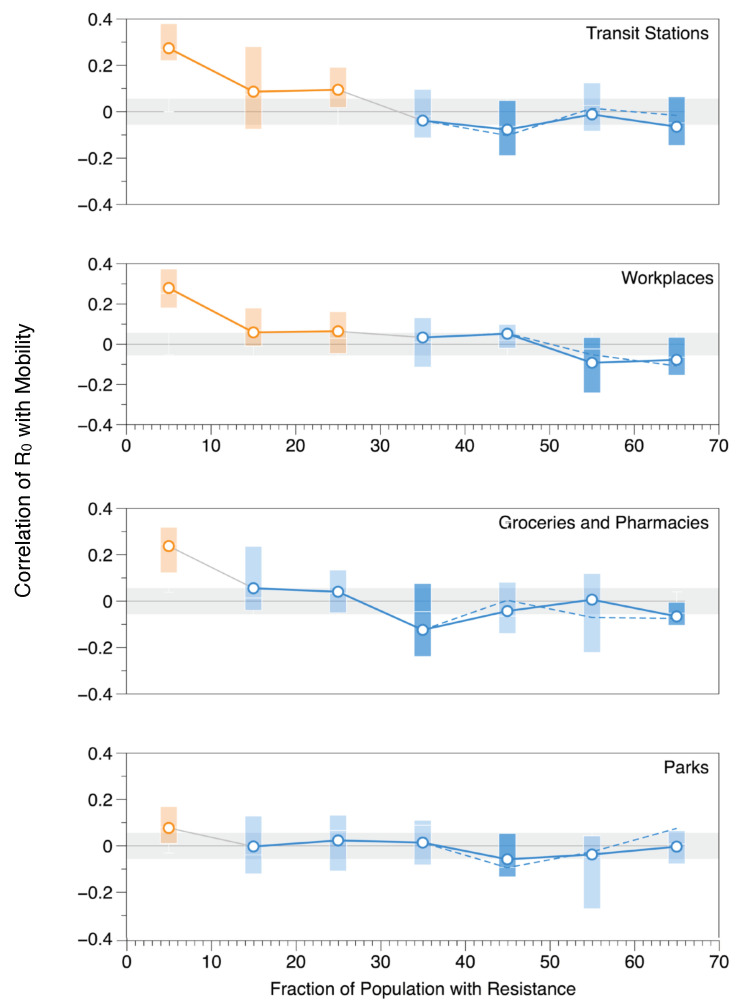
Correlation between COVID-19 transmission rates and mobility in transit stations, workplaces, groceries and pharmacies, and parks as a function of the fraction of the population with resistance, FR. The solid line only includes pre-Delta variants; the dashed line represents all variants, including Delta. Points in which there is a clear correlation are orange; points where the correlation is indistinguishable from zero (suggesting that mobility-restricting measures are no longer effective) are blue; grey regions indicate +/− the standard error in the mean.

**Table 1 ijerph-20-05343-t001:** Datasets Overview.

Dataset	Measurement Rate	Countries Coverage	Temporal Coverage
COVID-19 Measurements	Daily	231	Jan 2020—present
Google Mobility	Daily	123	Feb 2020—present
COVID-19 Variants	Two Weeks	110	May 2020—present

**Table 2 ijerph-20-05343-t002:** Country Multipliers, *m*.

Country	Multiplier	Source
US & Canada	4	[36]
Israel	1.5	[37]
France	7.94	[38,39]
Argentina	8.1	[40]
Uruguay	4	No Reference (used the US multiplier)
European countries	4	[38]

**Table 3 ijerph-20-05343-t003:** Correlation Mean Thresholds.

Mean	Color	Restriction Recommendation
>0.056 = standard error in the mean	Orange	Mobility and growth rate are positively correlated; may be helpful to restrict mobility
−0.056 and 0.056	Light blue	No correlation between mobility and growth rate; restricting mobility has little impact
<−0.056	Darker blue	Negative correlation between mobility and growth rate; restricting mobility potentially has an adverse effect.

**Table 4 ijerph-20-05343-t004:** Mobility-Restricting Measures for POIs.

POI	FR Need	Specification
Retail and Recreation	25–30%	Closed, crowded, stay a long time
Transit Station	25–30%	Closed, crowded
Workplaces	25–30%	Closed, crowded, stay a long time
Grocery and Pharmacy	10%	Closed but people just pick and go—might vary based on size
Parks	5–10%	Open and more social distancing

## Data Availability

Our World in Data [31]—Google provides a publicly available mobility dataset related to the pandemic [32]—Our World in Data shared a COVID-19 variants dataset [33].

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
