# Peer review of "The Effectiveness of Mobility Restrictions on Controlling the Spread of COVID-19 in a Resistant Population"

_ijerph, 2023, doi:10.3390/ijerph20075343_

Round 1

Reviewer 1 Report

The study is well-written and interesting in terms of policy implications.

LL 84-85 – the sentence ends in the midst; there is a "whereas...." but no continuation.

It seems reference #30 should come after reporting on 27 and 28. #29 is different in adding the mobility variable.

In presenting the gap in the literature, the authors forgot reference #29 which did examine both vaccination and mobility. The authors seem to be doing something additional – taking into account POI, the Delta variant, and the % of population recovered from COVID.

I need to say that I am unfamiliar  with the methodology used to calculate the growth rate of cases nor with estimation involving the Delta variant.

Figure 6 is unclear. Please include the meaning of the X and Y axis.

The discussion is very short. It is short of limitations & strengths of the study, and policy implications.

Misc.

Typo in Czechia; should be Czech republic.

Reviewer 2 Report

This paper analyzed the effects of mobility restriction measures on resistant populations using the COVID-19 dataset, the Google mobile dataset, and the COVID-19 variant. The conclusions obtained in this paper are reliable. However, there are still two doubts that need to be answered.

1. Does the difference in the time of immunity formation (after vaccination) in the fraction of resistance have an effect on the statistical results of this paper.

2. Whether the time scale of the data statistics (e.g., days, weeks) will affect the conclusions of this paper.
